# PIP5Kγ Mediates PI(4,5)P2/Merlin/LATS1 Signaling Activation and Interplays with Hsc70 in Hippo–YAP Pathway Regulation

**DOI:** 10.3390/ijms241914786

**Published:** 2023-09-30

**Authors:** Duong Duy Thai Le, Truc Phan Hoang Le, Sang Yoon Lee

**Affiliations:** 1Department of Biomedical Sciences, Ajou University Graduate School of Medicine, Suwon 16499, Gyeonggi-do, Republic of Korea; derek97le@ajou.ac.kr (D.D.T.L.); lphtruc91@ajou.ac.kr (T.P.H.L.); 2Institute of Medical Science, Ajou University School of Medicine, Suwon 16499, Gyeonggi-do, Republic of Korea

**Keywords:** PIP5Kγ, Hippo–YAP pathway, PI(4,5)P2, Merlin, LATS1, Hsc70, plasma membrane

## Abstract

The type I phosphatidylinositol 4-phosphate 5-kinase (PIP5K) family produces the critical lipid regulator phosphatidylinositol 4,5-bisphosphate (PI(4,5)P2) in the plasma membrane (PM). Here, we investigated the potential role of PIP5Kγ, a PIP5K isoform, in the Hippo pathway. The ectopic expression of PIP5Kγ87 or PIP5Kγ90, two major PIP5Kγ splice variants, activated large tumor suppressor kinase 1 (LATS1) and inhibited Yes-associated protein (YAP), whereas PIP5Kγ knockdown yielded opposite effects. The regulatory effects of PIP5Kγ were dependent on its catalytic activity and the presence of Merlin and LATS1. PIP5Kγ knockdown weakened the restoration of YAP phosphorylation upon stimulation with epidermal growth factor or lysophosphatidic acid. We further found that PIP5Kγ90 bound to the Merlin’s band 4.1/ezrin/radixin/moesin (FERM) domain, forming a complex with PI(4,5)P2 and LATS1 at the PM. Notably, PIP5Kγ90, but not its kinase-deficient mutant, potentiated Merlin–LATS1 interaction and recruited LATS1 to the PM. Consistently, PIP5Kγ knockdown or inhibitor (UNC3230) enhanced colony formation in carcinoma cell lines YAP-dependently. In addition, PIP5Kγ90 interacted with heat shock cognate 71-kDa protein (Hsc70), which also contributed to Hippo pathway activation. Collectively, our results suggest that PIP5Kγ regulates the Hippo–YAP pathway by forming a functional complex with Merlin and LATS1 at the PI(4,5)P2-rich PM and via interplay with Hsc70.

## 1. Introduction

The evolutionarily conserved Hippo pathway plays a critical role in maintaining normal tissue development by mediating cell contact-based inhibitory signaling [1,2,3]. In the Hippo pathway, large tumor suppressor kinase 1/2 (LATS1/2), which is phosphorylated and activated by mammalian Ste20-like kinase 1/2 (MST1/2), further phosphorylates Yes-associated protein (YAP) and transcriptional coactivator with PDZ-binding motif (TAZ) [2,3,4], mediating their inhibition via cytosolic retention through 14-3-3 protein binding or via ubiquitin-mediated proteasomal degradation [1,5]. Upon Hippo pathway inactivation, activated YAP and TAZ bind to transcription factor TEAD1–4 in the nucleus, inducing the expression of their target genes such as connective tissue growth factor (*CTGF*), cysteine-rich angiogenic inducer 61 (*CYR61*), and ankyrin repeat domain-containing protein 1 (*ANKRD1*) [4,6].

The neurofibromatosis type 2 (*NF2*) gene product, Merlin, is an important tumor suppressor that acts upstream of LATS1 to negatively regulate YAP in the Hippo pathway [7,8,9]. Merlin links the plasma membrane (PM) to the actin cytoskeleton like the ezrin/radixin/moesin (ERM) family proteins, plays a crucial role in regulating cell–cell junction and adhesion, and exerts cell contact-dependent anti-proliferative effects [7,10,11]. Merlin is activated upon binding to the membrane lipid phosphatidylinositol 4,5-bisphosphate (PI(4,5)P2) through its N-terminal band 4.1/ezrin/radixin/moesin (FERM) domain [12,13]. In its unbound state, the Merlin FERM domain interacts with its C-terminal domain, adopting an auto-inhibitory form that is released upon PI(4,5)P2 binding [14,15]. Notably, this conformational change induced by PI(4,5)P2 significantly increases Merlin binding affinity for LATS1 [12,16].

PI(4,5)P2 is a minor acidic phosphoinositide, the phosphorylated derivative of phosphatidylinositol, which critically regulates many cellular events such as membrane signaling, remodeling, and trafficking in the PM [17,18,19]. The recognition of PI(4,5)P2 by specific binding proteins affects their localization, activity, conformation, and interactions [19,20,21]. In mammals, PI(4,5)P2 is generated mainly by members of the type I phosphatidylinositol 4-phosphate 5-kinase (PIP5K) family, comprising PIP5Kα, PIP5Kβ, and PIP5Kγ, which catalyze the phosphorylation of phosphatidylinositol 4-phosphate substrate [22,23,24]. PIP5Kγ has two splice variants, PIP5Kγ87 (640 amino acids (aa), 87 kDa) and PIP5Kγ90 (668 aa, 90 kDa), reflecting the absence or presence of the C-terminal 28 aa, respectively [23,25].

The targeting of Merlin and LATS1 to the PM is critical for Hippo pathway activation [7,26,27]. Merlin recruits LATS1 to the PM, which, in turn, promotes its phosphorylation by MST1/2 [8]. It was reported that PI(4,5)P2, generated by PIP5Kα or PIP5Kγ, mediated the PM recruitment of Merlin [13,28]. However, the potential role of PIP5Kγ in the Hippo pathway was not fully previously characterized. In this study, we aimed to clarify how PIP5Kγ-dependent PI(4,5)P2 production coordinates Merlin/LATS1/YAP subcellular localization and signaling. We investigated the underlying mechanism by which PIP5Kγ regulates the Hippo pathway using genetic manipulations and ligand-induced Hippo signaling condition in human embryonic kidney and carcinoma cell lines. Here, we addressed the previously unexplored potential interaction and PM association of PIP5Kγ with Merlin and LATS1. In addition, we tested a possible implication of an interplay of PIP5Kγ with the heat shock cognate 71-kDa protein (Hsc70) in the regulation of the Hippo–YAP pathway.

## 2. Results

### 2.1. PIP5Kγ Has a Regulatory Role in the Hippo–YAP Pathway

To evaluate the potential role of PIP5Kγ in the Hippo pathway, we examined the effects of its overexpression on stimulatory LATS1 phosphorylation at serine 909 and inhibitory YAP phosphorylation at serine 127 using Western blotting (WB) analysis [2,29]. The transfection of human embryonic kidney 293 (HEK293) cells with HA-PIP5Kγ87 or HA-PIP5Kγ90 markedly increased LATS1 and YAP phosphorylation levels compared with those following control vector transfection (Figure 1A and Appendix A). Subcellular fractionation analysis showed that cytosolic YAP increased, and nuclear YAP decreased, following transfection with HA-PIP5Kγ87 or HA-PIP5Kγ90 (Figure 1B). Consistent with these results, the induction of YAP target genes, *CTGF*, *CYR61*, and *ANKRD1*, analyzed using quantitative real-time reverse transcription–polymerase chain reaction (qRT-PCR), was significantly reduced in HA-PIP5Kγ87- and HA-PIP5Kγ90-transfected cells compared with that in control vector-transfected cells (Figure 1C). Conversely, small interfering RNA (siRNA)-mediated PIP5Kγ knockdown in HEK293 cells or HeLa cells, which reduced both PIP5Kγ protein and PI(4,5)P2 levels, led to a notable decrease in LATS1 and YAP phosphorylation levels compared with those following control knockdown (Figure 1D,E and Appendix A). *CTGF*, *CYR61*, and *ANKRD1* expression levels were also higher in PIP5Kγ knockdown cells than those in control cells (Figure 1F). Conversely, adding back HA-PIP5Kγ87 or HA-PIP5Kγ90 to the PIP5Kγ knockdown cells increased LATS1 and YAP phosphorylation levels (Figure 1G and Appendix A).

Consistent with previous findings [28], we found that PIP5Kα overexpression or knockout (KO) regulated LATS1 and YAP in a similar manner (Appendix A). Moreover, overexpression (Appendix A) or genetic ablation (Appendix A) of both PIP5Kα and PIP5Kγ showed additive effects on LATS1 and YAP phosphorylation levels (Appendix A), and *CTGF*- and *CYR61*-inducing YAP transcriptional activity (Appendix A), suggesting that PIP5Kα and PIP5Kγ individually contribute to Hippo pathway activation by mediating LATS1 stimulation and YAP inhibition.

### 2.2. Regulatory Effects of PIP5Kγ90 on LATS1 and YAP Are Dependent on Its Catalytic Activity

We examined whether PIP5Kγ catalytic activity is involved in its regulation of LATS1 and YAP by comparing the effects of green fluorescent protein (GFP)-tagged kinase-dead (KD) PIP5Kγ90 (K407N, K408N) and wild-type (WT) forms. The indicated lysine residues are located in the kinase homology domain of the type I PIP5K family and mediate their catalytic function [30,31]. Unlike GFP-PIP5Kγ90 WT overexpression, GFP-PIP5Kγ90 KD overexpression did not effectively increase the phosphorylation levels of LATS1 and YAP (Figure 2A and Appendix A). GFP-PIP5Kγ90 KD was less effective in downregulating *CTGF*, *CYR61*, and *ANKRD1* induction than GFP-PIP5Kγ90 WT (Figure 2B). In accordance with these results, GFP-PIP5Kγ90 WT, but not the KD mutant, prevented the nuclear enrichment of YAP (Figure 2C). As expected, GFP-PIP5Kγ90 WT increased PI(4,5)P2 levels but GFP-PIP5Kγ90 KD did not (Figure 2D). These results suggest that PIP5Kγ modulates the Hippo–YAP pathway through its product PI(4,5)P2.

### 2.3. PIP5Kγ Mediates Restored YAP Phosphorylation in Response to EGF and LPA Stimulation

The Hippo signalings mediate cellular responses to various environmental cues such as serum starvation and stimulation [29]. Here, we tested whether PIP5Kγ could account for YAP phosphorylation upon epidermal growth factor (EGF) stimulation [32]. YAP phosphorylation in control knockdown cells was downregulated 30 min after EGF stimulation, followed by a recovery at 3 h (Figure 3A). In contrast, however, YAP phosphorylation levels remained relatively low during the 3 h period in PIP5Kγ knockdown cells (Figure 3A). Under the same conditions, we measured PI(4,5)P2 levels using its reporter construct, GFP-fused pleckstrin homology (PH) domain of phospholipase Cδ (PLCδ). As the PH domain preferentially binds to PM PI(4,5)P2, the PM (high PI(4,5)P2 levels) or cytoplasmic (low PI(4,5)P2 levels) localization of the transfected GFP-PLCδ-PH protein reflects the relative abundance of PI(4,5)P2 [33,34]. We observed that PI(4,5)P2 levels also underwent an initial decline at 30 min and then a rise at 3 h in EGF-stimulated control cells, but remained at relatively low levels in PIP5Kγ-knockdown cells (Figure 3B and Appendix A). When time-dependent effects of lysophosphatidic acid (LPA) stimulation on YAP phosphorylation were examined [29,32], the restoration of YAP phosphorylation at 2 h was relatively weak in PIP5Kγ-knockdown cells compared with that in control cells (Figure 3C), similar to the results obtained from EGF stimulation.

### 2.4. Merlin Binds to PIP5Kγ and Mediates Interaction of PIP5Kγ90 with LATS1

Next, we examined possible protein interactions of PIP5Kγ with Merlin and LATS1 using co-transfection and immunoprecipitation (IP) analysis. Myc-PIP5Kγ87 and Myc-PIP5Kγ90 coprecipitated with HA-Merlin (Figure 4A) and HA-LATS1 (Figure 4B). Endogenous LATS1 and Merlin coprecipitated with FLAG-PIP5Kγ90 (Figure 4C). Similarly, HA-IP products from HA-Merlin- or HA-LATS1-transfected cells contained endogenous PIP5Kγ (Figure 4D,E). GFP-Merlin and HA-LATS1 were present in the FLAG-IP products upon co-transfection with FLAG-PIP5Kγ90 (Figure 4F). FLAG-PIP5Kγ90 interacted with HA-Merlin under both control and LATS1 knockdown conditions (Figure 4G). However, Merlin KO abrogated the interaction of FLAG-PIP5Kγ90 with HA-LATS1 (Figure 4H).

### 2.5. PIP5Kγ90 Binds to the FERM Domain of Merlin and Modulates Merlin–LATS1 and YAP–TEAD4 Interactions

We examined possible interactions between the Merlin FERM domain and PIP5Kγ90. Both FLAG-Merlin WT and the FLAG-Merlin FERM domain strongly interacted with HA-PIP5Kγ90 (Figure 5A). To identify the binding motif in the Merlin FERM domain (18–312 aa), we next tested its F1 (18–98 aa), F2 (111–213 aa), and F3 (221–312 aa) subdomains [35]. The GFP-F3 subdomain exhibited marked interaction with HA-PIP5Kγ90 at levels comparable to the GFP-FERM domain (Figure 5B). Moreover, Myc-PIP5Kγ90 strengthened the HA-LATS1 interaction with GFP-Merlin (Figure 5C). Conversely, HA-PIP5Kγ90 weakened the interaction between FLAG-YAP1 and Myc-TEAD4 (Figure 5D). These results suggest that PIP5Kγ90 binds to the Merlin FERM domain, specifically to the F3 subdomain, and directs Hippo protein interactions leading to YAP repression.

### 2.6. PI(4,5)P2 Produced by PIP5Kγ90 Induces LATS1 Translocation to Merlin-Rich PM Regions

We next addressed the potential effects of a PIP5Kγ90 WT and KD mutant on Merlin and LATS1 localization through the confocal imaging of co-transfected HeLa cells. Both GFP-PIP5Kγ90 WT and KD mutant forms co-localized with HA-Merlin without altering its PM localization (Figure 6A). However, GFP-PIP5Kγ90 KD reduced the PM localization of HA-LATS1, in contrast to its strong PM co-localization with GFP-PIP5Kγ90 WT (Figure 6B). Similarly, Myc-LATS1 co-localized with GFP-PIP5Kγ90 WT and HA-Merlin in the PM but was dislocated from the PM upon GFP-PIP5Kγ90 KD co-transfection (Figure 6C). Notably, FLAG-Merlin interacted with GFP-PIP5Kγ90 WT and endogenous LATS1 (Figure 6D) but lost its LATS1 binding affinity upon GFP-PIP5Kγ90 KD co-transfection, although it still interacted with the KD mutant (Figure 6D), consistent with the imaging results shown in Figure 6C.

To visualize these proteins together with PI(4,5)P2, we used the monomeric red fluorescent protein (mRFP)-Tubby as a PI(4,5)P2-specific probe, which also selectively binds to PI(4,5)P2; thus it undergoes translocation between the PM and the cytoplasm depending on PM PI(4,5)P2 levels [33,36]. In GFP-PIP5Kγ90 WT-transfected cells, HA-Merlin or HA-LATS1 co-localized with mRFP-Tubby at the PM (Figure 6E,F). Similar to the results shown in Figure 6A–C, upon GFP-PIP5Kγ90 KD co-transfection, HA-Merlin localized at the PM (Figure 6E), whereas mRFP-Tubby and HA-LATS1 exhibited cytosolic distribution (Figure 6F). Collectively, these results suggest that the PIP5Kγ90-dependent PI(4,5)P2 pool mediates LATS1 recruitment to the local PM sites enriched in Merlin, but does not substantively affect the PM localization of Merlin (Figure 6G).

### 2.7. Merlin and LATS1 Are Necessary for Hippo–YAP Pathway Regulation by PIP5Kγ

We then examined effects of PIP5Kγ on the Hippo pathway in Merlin- or LATS1-deficient HEK293A cells to test whether PIP5Kγ-mediated Hippo pathway regulation is dependent on Merlin and LATS1. HA-PIP5Kγ87 or HA-PIP5Kγ90 transfection led to increases in LATS1 and YAP phosphorylation levels in WT HEK293A cells, whereas these effects were attenuated in Merlin KO HEK293A cells (Figure 7A and Appendix A). Under the transfection conditions, *CTGF*, *CYR61*, and *ANKRD1* expression levels were decreased in WT cells, but remained relatively high in Merlin KO cells (Figure 7B). Similarly, LATS1/2 KO abrogated the upregulating effects of HA-PIP5Kγ87 and HA-PIP5Kγ90 on YAP phosphorylation (Figure 7C and Appendix A) and their downregulating effects on the YAP target-gene induction (Figure 7D). Consistent with these results, transfection with GFP-PIP5Kγ87 or GFP-PIP5Kγ90 prevented YAP nuclear enrichment in WT cells, albeit to a much lesser degree in Merlin KO or LATS1/2 KO cells (Figure 7E–H). These results suggest that Merlin and LATS1 play a role as major downstream effectors in the PIP5Kγ-mediated Hippo–YAP pathway regulation.

### 2.8. PIP5Kγ Has the Potential to Suppress YAP-Dependent Cell Proliferation

Considering the involvement of the Hippo–YAP pathway in tumorigenesis, we then evaluated the potential role of PIP5Kγ in cancer by mining open bioinformatics datasets. According to the Human Protein Atlas (https://www.proteinatlas.org/ (accessed on 7 April 2022)) [37], PIP5Kγ is recognized as a favorable prognostic marker in cervical, endometrial, pancreatic, and renal cancer patients (Appendix A). Similarly, the Kaplan–Meier Plotter (https://kmplot.com/ (accessed on 11 August 2022)) shows that high PIP5Kγ expression levels positively correlate with survival probability in head and neck cancer patients (Appendix A) [38]. In accordance with these, the Gene Expression Profiling Interactive Analysis (http://gepia2.cancer-pku.cn/ (accessed on 6 May 2022)) [39] indicates that PIP5Kγ gene expression levels are relatively low in various human cancers, including cervical cancer, compared to those in normal samples (Appendix A).

In these contexts, we further tested effects of PIP5Kγ on cancer cell proliferation. Treatment of HeLa (human cervical carcinoma) cells with UNC3230, a PIP5Kγ-specific inhibitor [40], reduced PI(4,5)P2 levels (Figure 8A). UNC3230 enhanced HeLa cell colony formation, which was prevented by its co-treatment with verteporfin, an inhibitor of YAP [41] (Figure 8B,C). In turn, siRNA-mediated PIP5Kγ knockdown enhanced colony formation in FaDu cells (a cell-line model of human head and neck squamous cell carcinoma) (Figure 8D). Treatment of HeLa cells with verteporfin significantly blocked colony formation increased by PIP5Kγ siRNA (Figure 8E,F). Likewise, PIP5Kγ siRNA and verteporfin had opposite effects on proliferation-related cell viability (Figure 8G). These results suggest an antagonizing effect of PIP5Kγ on YAP-driven cell proliferation.

### 2.9. Interplay between PIP5Kγ90 and Hsc70 Leads to Activation of the Hippo Pathway

Using the free available KFERQ finder V0.8 website (https://rshine.einsteinmed.edu/ (accessed on 13 October 2022)) [42], we noticed that PIP5Kγ has a canonical KFERQ-like motif (QDFRF, 129–133 aa) that can bind to Hsc70 (also termed HSPA8), which is required for the chaperone-mediated autophagy (CMA) [43]. Such Hsc70-binding KFERQ-like motifs are not found in PIP5Kα and PIP5Kβ, based on the web search. Here, we tested whether PIP5Kγ could be a CMA substrate via the Hsc70 binding and this potential interaction could affect the PIP5Kγ-mediated Hippo pathway regulation. Upon transfection, FLAG-PIP5Kγ90 coprecipitated with V5-Hsc70 (Figure 9A) and endogenous Hsc70 (Figure 9B). PIP5Kγ IP products also showed endogenous PIP5Kγ–Hsc70 interaction (Figure 9C). However, transfected V5-Hsc70 did not induce the protein degradation of PIP5Kγ (Figure 9D). Consistently, treatments with atypical retinoid 7 (AR7) and 6-aminonicotinamide (6AN), which are known as chemical activators of CMA [44,45], had no degrading effects on PIP5Kγ protein levels (Appendix A).

Next, we examined PIP5Kγ–Hsc70 interaction in the absence and presence of Merlin or LATS1 using their KO cells. Comparable levels of FLAG-PIP5Kγ90 interaction with V5-Hsc70 were still detected in Merlin KO (Figure 9E) and LATS1/2 KO (Figure 9F) HEK293A cells. Notably, we found that transfected V5-Hsc70 further augmented interactions of FLAG-PIP5Kγ90 with HA-Merlin and endogenous LATS1, as well as YAP phosphorylation levels (Figure 9G). In addition, Hsc70 knockdown by its siRNA reduced phosphorylation levels of LATS1 and YAP in both basal (control vector-transfected) and HA-PIP5Kγ90-transfected conditions compared with those in the control knockdown condition (Figure 9H and Appendix A). Likewise, we tested the effects of transfected V5-Hsc70 on those phosphorylations in control and PIP5Kγ-knockdown cells. Interestingly, V5-Hsc70 overexpression elevated LATS1 and YAP phosphorylation levels in control knockdown cells, and these changes were attenuated in PIP5Kγ-knockdown cells (Figure 9I and Appendix A). These findings suggest a supporting role of the binding and interplay between PIP5Kγ and Hsc70 in Hippo pathway activation.

## 3. Discussion

In this study, we identified the precise role of PIP5Kγ in regulating the Hippo–YAP pathway. Our results showed that both PIP5Kγ87 and PIP5Kγ90 ultimately activated LATS1 and inhibited YAP activity in a similar manner. Merlin and LATS1 were required for the regulation of the Hippo–YAP pathway by the two PIP5Kγ splice variants. The regulatory effects of PIP5Kγ90 on LATS1 and YAP are PI(4,5)P2-dependent; moreover, because the K407 and K408 residues mutated in the PIP5Kγ90 KD form are conserved in PIP5Kγ87, it is likely that the PIP5Kγ87 regulatory effects are also PI(4,5)P2-dependent. Our results support a working model in which PIP5Kγ-dependent PI(4,5)P2 generation mediates the interaction and PM enrichment of LATS1 with Merlin, which induces LATS1 activation and subsequent YAP inhibition.

We showed the positive correlation between changes in PI(4,5)P2 levels and changes in YAP phosphorylation levels in the time course of EGF and LPA stimulation. PIP5Kγ knockdown resulted in aberrantly lowered YAP phosphorylation during the stimulation, supporting the idea that PIP5Kγ can serve as an endogenous regulator of the Hippo–YAP signaling pathway through PI(4,5)P2 generation. Growth factor stimulation is accompanied by an activation of PI(4,5)P2-sensitive PLC, such as PLCγ1, that hydrolyzes PI(4,5)P2 [46]. LPA ligation into its G-protein-coupled receptor can activate the downstream effector PLC [47,48]. Thus, it is likely that a rapid decline in PI(4,5)P2 upon EGF and LPA stimulation results from PLC activation, and then PIP5Kγ subsequently mediates the replenishment of PI(4,5)P2.

Moreover, we found that the F3 subdomain of the Merlin FERM domain binds to PIP5Kγ90. As Merlin also interacted with PIP5Kγ87, it is plausible that a common, albeit yet-unidentified, motif between PIP5Kγ87 and PIP5Kγ90 mediates Merlin binding. We also observed that LATS1 interacts with PIP5Kγ87 and PIP5Kγ90. Accordingly, we identified a ternary protein complex comprising PIP5Kγ90, Merlin, and LATS1. Our data suggest that whereas PIP5Kγ90 directly interacts with Merlin independently of LATS1, the PIP5Kγ90–LATS1 interaction is indirectly mediated through Merlin. As LATS1 binds to the Merlin FERM F2 subdomain [8,49], the Merlin FERM domain may serve as a platform for the formation of a functional complex with PI(4,5)P2, PIP5Kγ, and LATS1. We previously found such a ternary complex among PIP5Kα, Merlin, and LATS1 but, in case of PIP5Kα, it specifically bound to the F1 subdomain within the Merlin FERM domain [34]. These suggest that PIP5Kα and PIP5Kγ can bind to Merlin in a non-competitive way, which is further supported by the results of their additive effects on Hippo pathway activation. Additionally, our data suggest that PIP5Kγ90 may negatively regulate the interaction between YAP and TEAD4.

Merlin plays an important role in Hippo pathway activation by targeting LATS1 to the PM, leading to enhanced MST1/2-mediated LATS1 phosphorylation [8]. Thus, it is reasonable that PIP5Kγ-generated PI(4,5)P2, at least in part, contributes to the PM co-enrichment of LATS1 with Merlin, likely by inducing Merlin conformational activation [12,14,15]. Moreover, we observed that PIP5Kγ90, but not its KD mutant, intensifies the interaction between Merlin and LATS1. Given that PI(4,5)P2 binding to Merlin promotes Merlin–LATS1 interaction [12,16], our results support the model that the PIP5Kγ–Merlin interaction facilitates the PI(4,5)P2-dependent Merlin activation owing to their spatial proximity, which, in turn, enhances the ability of Merlin to interact with LATS1 and promote its activation.

The PIP5K isoforms perform distinct functions but, often, also show overlapping roles [24,50]. PIP5Kα and PIP5Kγ play critical roles in osmotic stress-induced Hippo pathway activation by mediating the PI(4,5)P2-dependent PM recruitment of Merlin [28]. In a similar context, we recently reported that PIP5Kα-dependent PI(4,5)P2 production was involved in Hippo pathway activation via Merlin and LATS1 in different cell densities and serum starvation/stimulation conditions [34]. However, according to our previous results, PIP5Kα induced the cytosol-to-PM translocation of LATS1 without affecting the PM location of Merlin [34]. In this present study, our results support the idea that PI(4,5)P2 produced by PIP5Kγ also functions as an activator of Merlin rather than a mediator of its PM recruitment. Based on these findings observed in our experimental conditions, we propose that PIP5Kα and PIP5Kγ play a similar role in Hippo–YAP pathway regulation in that both PIP5K isoforms mediated the Merlin activation and PM recruitment of LATS1 in a PI(4,5)P2-dependent manner. PI(4,5)P2 is also generated by the type II phosphatidylinositol 5-phosphate 4-kinase (PIP4K) family members to phosphorylate phosphatidylinositol 5-phosphate substrate [50]. PI(4,5)P2-specific PLCγ1 to degrade PI(4,5)P2 was engaged in the small G-protein RAP2-mediated control of the Hippo–YAP pathway [51]. Thus, besides PIP5Kα and PIP5Kγ, those PI(4,5)P2-metabolizing enzymes may also have the potential to regulate Merlin/LATS1/YAP signalings in the Hippo pathway.

The results of our bioinformatics analyses raise the possibility that PIP5Kγ is implicated in certain types of cancer such as cervical cancer and head and neck cancer. Our results showed an anti-proliferative effect of PIP5Kγ on HeLa and FaDu cells, the cell-line models of those cancers. In accordance with these results, PIP5Kγ knockdown exerted a proliferative effect in a YAP-dependent manner. As PIP5Kγ was physically associated with Merlin, our findings would imply that the Merlin-linked Hippo–YAP pathway is a relevant signaling target of PIP5Kγ in cancer. However, it remains to be determined whether the PIP5Kγ/Merlin/LATS1 signaling axis plays a functional role in tumor progression. Alternatively, PIP5Kγ90 promoted the well-established oncogenic phosphatidylinositol 3-kinase (PI3K)/Akt signaling pathway by serving the PI3K-catalyzed phosphatidylinositol 3,4,5-trisphosphate generation [52]. In this regard, PIP5Kγ may serve as a physiological regulator of cell growth and proliferation by switching downstream signaling pathways between Merlin/LATS1 and PI3K/Akt, depending on certain changes in the extracellular or intracellular microenvironment.

Based on the presence of the Hsc70-binding KFERQ-like motif (QDFRF) in PIP5Kγ and its interaction with Hsc70, we initially hypothesized that PIP5Kγ could be subject to degradation by CMA. However, both Hsc70 overexpression and treatments with CMA activators did not induce PIP5Kγ degradation. Currently, it remains unclear whether PIP5Kγ binds to Hsc70 via the QDFRF motif without being degraded, or via other motif(s) except the QDFRF motif. This protein interaction appeared independent of Merlin and LATS1. Intriguingly, we found that Hsc70 reinforced PIP5Kγ90 interactions with Merlin and LATS1 and the inhibitory YAP phosphorylation. These imply that Hsc70 may be associated with the PIP5Kγ/Merlin/LATS1 signaling axis in a cooperative way, although more details about how Hsc70 promotes the interaction of PIP5Kγ90 with Merlin and LATS1 need to be clarified. A recent study demonstrated that YAP1 is a substrate to undergo degradation via CMA through KFERQ-like motif-mediated Hsc70 binding [53]. However, the role of Hsc70 in Hippo pathway regulation largely remains elusive. Here, we observed that Hsc70 as well as PIP5Kγ led to increases in the LATS1 S909 and YAP S127 phosphorylations. In addition, our results showed that increased LATS1 and YAP phosphorylation levels by PIP5Kγ90 and Hsc70 were strikingly reduced upon Hsc70 and PIP5Kγ knockdown, respectively. These results further raise a possibility that an interplay with PIP5Kγ may be one of mechanisms by which Hsc70 functions as a regulator of the Hippo–YAP pathway.

PIP5K enzymatic activity is controlled by multiple mechanisms that involve small G-proteins of the Rho family, ADP-ribosylation factor 6, phosphatidic acid, or phosphorylation modifications [24,54,55]. PIP5K expression and protein stability can also affect PI(4,5)P2 levels [33]. Various factors participate in Hippo–YAP pathway regulation depending on different environmental contexts, such as cell–cell contacts, cell polarity, cell density, and serum deprivation [2,56,57,58,59]. Thus, it is worthwhile to examine how PIP5Kγ activity, expression, and/or stability are distinctively modulated under specific Hippo signaling conditions. Overall, our results provide insights regarding the functional roles of PIP5Kγ and PI(4,5)P2 in Hippo–YAP pathway regulation.

## 4. Materials and Methods

### 4.1. Chemicals and Antibodies

Dulbecco’s modified Eagle’s medium (DMEM), minimum essential medium, EGF, LPA, verteporfin, crystal violet, poly-L-lysine, and anti-FLAG M2 Affinity agarose gel (A2220) were purchased from Sigma-Aldrich (St. Louis, MO, USA). Lipofectamine 2000, Lipofectamine RNAiMAX, Opti-MEM I, DAPI, goat serum, and Alexa Fluor-conjugated secondary antibodies were from Thermo Fisher Scientific (Waltham, MA, USA). UNC3230, AR7, and 6AN were purchased from MedChemExpress (Monmouth Junction, NJ, USA). Antibodies to α-tubulin (T5168), β-actin (A5316), vinculin (V4505), and FLAG-tag (F1804) were obtained from Sigma-Aldrich. Antibodies to HA-tag (#3724), Myc-tag (#2278), V5-tag (#13202), LATS1 (#9153), phospho-LATS1 Ser909 (#9157), YAP (#4912), phospho-YAP Ser127 (#4911), Merlin (#6995), and PIP5Kγ (#3296) were purchased from Cell Signaling Technology (Danvers, MA, USA). Antibodies against HA-tag (sc-7392), lamin B1 (sc-374015), Hsc70 (sc-7298), GFP (sc-9996), and GAPDH (sc-47724) were obtained from Santa Cruz Biotechnology (Dallas, TX, USA).

### 4.2. Expression Constructs

HA-, FLAG-, Myc-, or GFP-tagged PIP5Kγ87 and/or PIP5Kγ90 and FLAG-PIP5Kα have been described previously [36,55,60]. Myc-PIP5Kγ87 and GFP-PIP5Kγ90 mutant (K407N and K408N) were gifts from Prof. Toshiki Itoh (Kobe University, Kobe, Japan) and Prof. Pietro De Camilli (Yale University, New Haven, CT, USA), respectively. HA-LATS1 plasmid was gifted by Dr. Eunjeong Seo (OliPass Corporation, Yongin, Gyeonggi, Republic of Korea). FLAG-Merlin (#11623), HA-Merlin (#32836), GFP-Merlin (#84293), FLAG-Merlin FERM (#11625), Myc-LATS1 (#66851), FLAG-YAP1 (#66853), V5-Hsc70 (#19514), and Myc-TEAD4 (#24638) were purchased from Addgene (Cambridge, MA, USA). GFP-tagged plasmids of the Merlin FERM domain and its F1, F2, and F3 subdomains were generated using PCR with the GFP-Merlin template and ligated into the EcoRI–BamHI sites of pEGFP-C2 vector (GenBank Accession No.: U57606; BD Biosciences Clontech, Palo Alto, CA, USA) [34]. The generated recombinant plasmids were confirmed using DNA sequencing (Bionics, Seoul, Republic of Korea). All plasmids were purified using the Plasmid Midi Kit (Qiagen, Hilden, Germany).

### 4.3. Cell Culture

HEK293, HeLa, and FaDu cell lines were purchased from the American Type Culture Collection (Manassas, VA, USA). HEK293 and HeLa cells along with control and PIP5Kα KO HEK293 cells [33] were grown in DMEM supplemented with 10% fetal bovine serum (FBS) and penicillin/streptomycin (HyClone, Logan, UT, USA) at 37 °C in a humidified atmosphere (5% CO_2_, 95% air). FaDu cells were cultured in minimum essential medium supplemented with 10% FBS and penicillin/streptomycin. WT, LATS1/2 KO, and Merlin KO HEK293A cells were provided by Prof. Jung Soon Mo (Ajou University Graduate School of Medicine) and maintained in DMEM containing 10% FBS and the antibiotics. Cells were subcultured at 2- or 3-day intervals.

### 4.4. Transfection and Gene Knockdown

Cells were transiently transfected for 24 h with the indicated plasmids (0.25 μg/cm^2^) or corresponding empty vectors mixed with Lipofectamine 2000 in Opti-MEM I. For PIP5Kγ and Hsc70 knockdown, their siRNAs (sc-39137 and sc-29349, respectively, Santa Cruz Biotechnology) mixed with Lipofectamine RNAiMAX in Opti-MEM I were added to cells at a final 20 nM concentration for 48 h. Similarly, LATS1 siRNA targeting the sequence (5′-GAACCAAACUCUCAAACAAdTdT-3′) and a non-targeting siRNA (5′-UUCUCCGAACGUGUCACGUdTdT-3′) from Bioneer (Daejeon, Republic of Korea) were used for LATS1 and control knockdown, respectively [34].

### 4.5. WB and IP Analysis

WB and IP experiments were performed as previously described [33,60]. Briefly, cells were harvested in ice-cold lysis buffer (50 mM Tris pH 7.4, 150 mM NaCl, 1 mM egtazic acid, 5 mM NaF, 1 mM dithiothreitol, 1 mM Na_3_VO_4_, and 1% Triton X-100) containing protease and phosphatase inhibitor cocktails (GenDEPOT, Barker, TX, USA), and placed on ice for 15 min with occasional vortexing. Cell lysates were prepared using centrifugation (17,000× *g*, 25 min, 4 °C), and protein concentrations were measured using bicinchoninic acid protein assay reagents (Pierce, Rockford, IL, USA) and bovine serum albumin as a protein standard. For HA- and GFP-tagged protein IP, cell lysates (2.0 mg) were incubated with anti-HA antibody (3 μg) and anti-GFP antibody (4 μg), respectively, for 4 h at 4 °C, followed by incubation with protein A/G PLUS-Agarose IP reagent (30 μL; Santa Cruz Biotechnology) for an additional 2 h. Likewise, FLAG-tagged proteins were immunoprecipitated by mixing cell lysates with anti-FLAG M2 affinity gels (25 μL) for 4 h at 4 °C. In the case of endogenous PIP5Kγ IP, cell lysates (2.5 mg) were mixed with control IgG or anti-PIP5Kγ antibody (5 μg) for 6 h at 4 °C, then with protein A/G PLUS-Agarose IP reagent for 2 h. IP products were washed five times with the lysis buffer.

The resulting proteins in the cell lysates and IP products were separated using sodium dodecyl sulfate–polyacrylamide gel electrophoresis (SDS-PAGE) on 8–10% resolving gels and transferred to Immobilon-P polyvinylidene difluoride membranes (Merck Millipore, Billerica, MA, USA). After blocking with 5% non-fat dry milk in Tris-buffered saline containing 0.1% Tween-20 (TBST), the membranes were blotted with the indicated primary antibodies for 2 h at room temperature or overnight at 4 °C, then washed three times with TBST. Following further blotting with horseradish peroxidase-conjugated secondary antibodies (Jackson ImmunoResearch Laboratories, West Grove, PA, USA), target proteins were detected using SuperSignal™ West Pico PLUS chemiluminescent substrate (Thermo Fisher Scientific). GAPDH, β-actin, α-tubulin, or vinculin immunoblottings were included as a loading control.

### 4.6. qRT-PCR Analysis

Total RNA was purified using a Ribospin II RNA purification kit (GeneAll, Seoul, Republic of Korea) and cDNA was synthesized using ReverTra Ace™ qPCR RT Master Mix (Toyobo Company, Osaka, Japan). qRT-PCR was performed using a QuantStudio™ 3 Real-Time PCR System (Thermo Fisher Scientific) with TOPreal™ SYBR Green qPCR 2X PreMIX (Enzynomics, Daejeon, Republic of Korea). PCR samples containing specific target gene primers (Table 1) from Bioneer were prepared in triplicate. The following PCR reaction settings were used: a single cycle at 95 °C for 15 min, followed by 40 cycles with three steps each at 95 °C for 10 s, 60 °C for 15 s, and 72 °C for 30 s. mRNA levels were normalized to those of *GAPDH*; relative expression was determined using the 2^−ΔΔCt^ method.

### 4.7. Immunostaining and Cell Imaging

HeLa cells were seeded into 12-well plates containing 18 mm circular coverslips precoated with poly-L-lysine (0.5 mg/mL) approximately 12 h prior to transfection. One day post-transfection with the indicated plasmids as described above, cells were washed with phosphate-buffered saline (PBS), permeabilized with PBS containing 0.1% Triton X-100 for 15 min. Following blocking with PBS containing 10% bovine serum albumin and 5% goat serum for 30 min, transfected HA- or Myc-tagged proteins and endogenous YAP were immunostained with primary antibodies (1:100 dilution) for 2 h at 25 °C, and then with Alexa Fluor 350- and/or 594-conjugated secondary antibodies (1:200 dilution) for 1 h, as described previously [60]. Nuclei were stained with DAPI diluted in PBS for 5 min. The immunostained cells were washed with PBS between each staining step, then with distilled water. After the samples were mounted with Prolong Gold anti-fade reagent (Thermo Fisher Scientific), fluorescence images of immunostained proteins and transfected GFP- and mRFP-tagged proteins were captured using an LSM 710 confocal microscope (Carl Zeiss GmbH, Jena, Germany).

### 4.8. PI(4,5)P2 Visualization

PI(4,5)P2 was visualized through immunostaining with an anti-PI(4,5)P2 mouse IgM monoclonal antibody (Z-P045; Echelon Biosciences, Salt Lake City, UT, USA), followed by sequential incubation with biotinylated goat anti-mouse IgM secondary antibody (1:200 dilution; Jackson ImmunoResearch Laboratories) and Alexa Fluor 594-conjugated streptavidin (1:500 dilution; Thermo Fisher Scientific) [36,60]. Alternatively, PI(4,5)P2 imaging was performed using its fluorescent reporter constructs, mRFP-Tubby or GFP-PLCδ-PH as previously described [33,34,36]. PI(4,5)P2 immunofluorescence and GFP fluorescence intensities were measured using ImageJ (National Institutes of Health, Bethesda, MD, USA) and Zeiss ZEN 3.3 imaging software, respectively.

### 4.9. Subcellular Fractionation

Nuclear and cytosolic fractions were prepared using a Nuclear Extraction Kit (#2900, Merck Millipore) according to the supplier’s instructions. Enrichment of nuclear and cytosolic proteins was tested using lamin B1 and α-tubulin WB analysis, respectively.

### 4.10. Colony Formation and Cell Viability Assays

After seeding into 6-well plates (1000 cells/well), cells were repeatedly treated with UNC3230 (100 nM), verteporfin (1 μM), and/or DMSO as a vehicle control at 2 to 3 d intervals. Cells were treated with control or PIP5Kγ siRNA every 3 days under the indicated conditions. After 10 d, the cells were stained with 0.5% crystal violet following fixation with a mixture of acetic acid and methanol at 25 °C, as described previously [33]. Cell colonies were counted using ImageJ software. Cell viability was evaluated using an EZ-CYTOX assay kit (Daeil Lab Service, Seoul, Republic of Korea) according to the manufacturer’s protocol. Briefly, cells seeded in 96-well plates (10,000 cells/well) were treated with PIP5Kγ siRNA and/or verteporfin (1 μM) as described above. After 7 d, formation of the water-soluble tetrazolium salt was measured at 450 nm using a BioTek Synergy H1 Plate Reader (Agilent Technologies, Santa Clara, CA, USA).

### 4.11. Statistical Analysis

All experiments were independently repeated at least three times with similar results. The band intensities of LATS1 and YAP phosphorylation in the Western blots were measured using ImageJ software. Data shown in the graphs are the mean ± S.E.M. The *p* values  <  0.05 were regarded as statistically significant and were determined using GraphPad Prism 8 software (La Jolla, CA, USA). The unpaired Student’s *t*-test was used to compare two groups; one-way analysis of variance with Tukey’s multiple-comparison test was used for three or more groups.

## 5. Conclusions

In this study, we demonstrated that PIP5Kγ serves as an activator of the Hippo pathway via Merlin and LATS1. Mechanistically, our results suggest that PIP5Kγ generates PI(4,5)P2 at the PM; the lipid kinase and product then both bind to the Merlin FERM domain, allowing its activation, which then promotes the Merlin–LATS1 interaction and induces LATS1 recruitment to the PI(4,5)P2-enriched PM sites (Figure 10). Consequently, the PIP5Kγ-mediated Merlin-dependent LATS1 activation suppresses YAP, and thus YAP-associated cell proliferation. PIP5Kγ90 interaction with Hsc70 also contributes to Hippo pathway activation, likely through facilitating PIP5Kγ90–Merlin interaction. Further investigation to identify the potential implications of PIP5Kγ/PI(4,5)P2 in Merlin/LATS1 and Hsc70 signalings in various Hippo–YAP pathway-related physiological and pathological processes is, therefore, warranted.

## Figures and Tables

**Figure 1 ijms-24-14786-f001:**
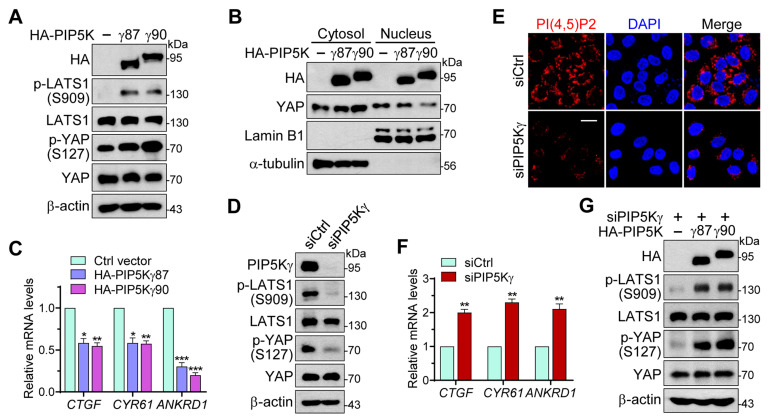
Effects of PIP5Kγ overexpression or knockdown on LATS1 and YAP. (**A**–**C**) HEK293 cells were transfected with control vector, HA-PIP5Kγ87, or HA-PIP5Kγ90. HEK293 (**D**,**F**) or HeLa (**E**) cells were transfected with control siRNA (siCtrl) or PIP5Kγ siRNA (siPIP5Kγ). (**G**) HA-PIP5Kγ87 or HA-PIP5Kγ90 was transfected into HEK293 cells pretreated with siPIP5Kγ. Cell lysates (**A**,**D**,**G**) and cytosolic and nuclear fractions (**B**) were analyzed using WB with the indicated antibodies. (**C**,**F**) Relative quantification of *CTGF*, *CYR61*, and *ANKRD1* mRNA levels analyzed by qRT-PCR (*n* = 3). Values in the graphs represent the mean ± S.E.M. * *p* < 0.05, ** *p* < 0.01, *** *p* < 0.001. (**E**) Confocal images of PI(4,5)P2 immunostaining. Cells were visualized using nuclear staining with 4′,6-diamidino-2-phenylindole (DAPI). Scale bar, 10 μm.

**Figure 2 ijms-24-14786-f002:**
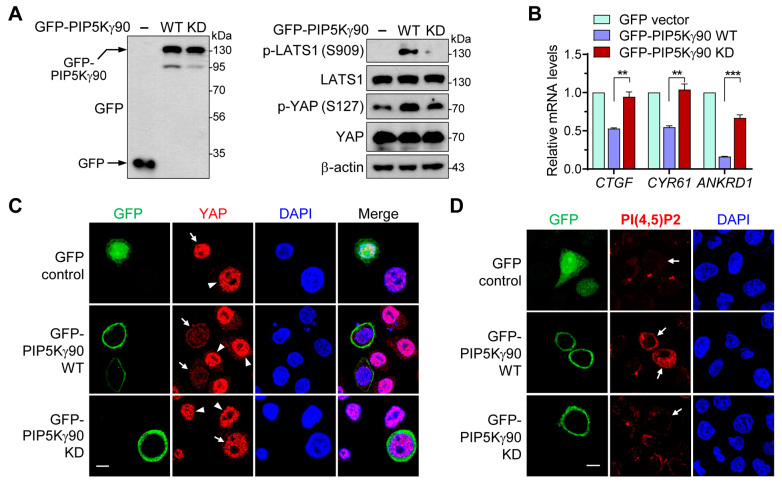
Effects of PIP5Kγ90 WT and its KD mutant on LATS1 and YAP. HEK293 (**A**,**B**) or HeLa (**C**,**D**) cells were transfected with GFP control vector, GFP-PIP5Kγ90 WT, or GFP-PIP5Kγ90 KD. (**A**) WB analysis of cell lysates with the indicated antibodies. Arrows indicate GFP or GFP-PIP5Kγ90. (**B**) Relative quantitation of *CTGF*, *CYR61*, and *ANKRD1* mRNA levels as analyzed using qRT-PCR (*n* = 3). Values in the graphs represent the mean ± S.E.M. ** *p* < 0.01, *** *p* < 0.001. Cells were immunostained with anti-YAP (**C**) or anti-PI(4,5)P2 (**D**) antibodies, and nuclei were stained with DAPI. Representative images of GFP, YAP, or PI(4,5)P2 immunofluorescence, and DAPI-stained nuclei were captured using confocal microscopy. The arrows and arrowheads indicate transfected and non-transfected cells, respectively. Scale bars, 10 μm.

**Figure 3 ijms-24-14786-f003:**
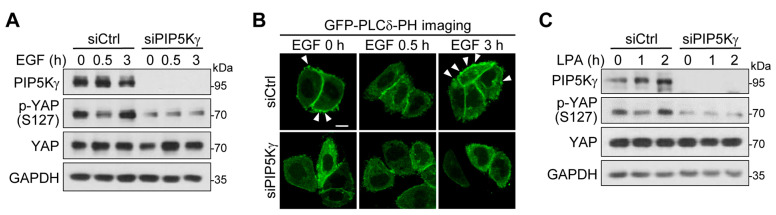
Effects of PIP5Kγ knockdown on YAP phosphorylation and/or PI(4,5)P2 levels upon EGF and LPA treatment. (**A**–**C**) HeLa cells were treated with control siRNA (siCtrl) or PIP5Kγ siRNA (siPIP5Kγ) for 24 h. Following serum starvation for an additional 16 h, cells were left untreated or treated with 50 ng/mL EGF (**A**) or 10 μM LPA (**C**) for the indicated times. (**A**,**C**) Resulting cell lysates were immunoblotted with the indicated antibodies. (**B**) One day after siRNA treatment, cells were transfected with a PI(4,5)P2-specific fluorescent probe, GFP-PLCδ-PH before EGF treatment. Representative GFP fluorescence images were obtained using confocal microscopy. Arrowheads represent PM enrichment sites of the transfected protein, indicative of relatively high PI(4,5)P2 levels. Scale bar, 10 μm.

**Figure 4 ijms-24-14786-f004:**
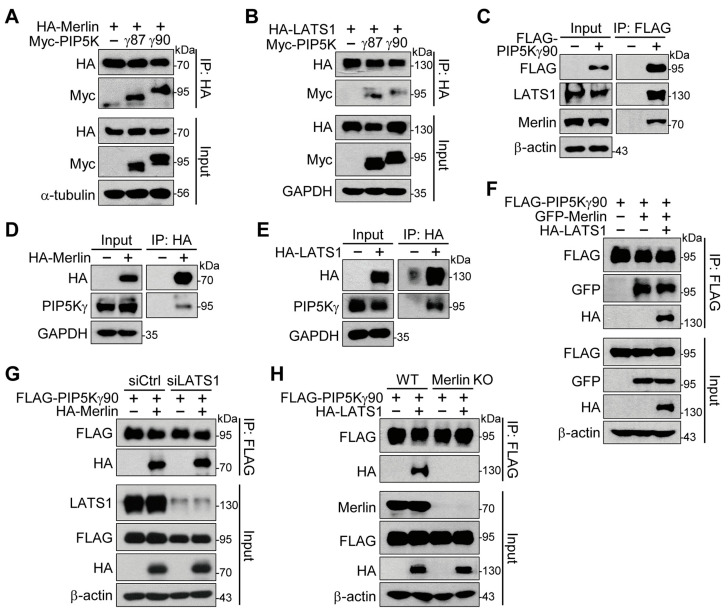
Interaction of PIP5Kγ90 with Merlin and LATS1. HEK293 cells (**A**–**G**) or WT and Merlin KO HEK293A cells (**H**) were transfected in the absence and presence of the indicated expression plasmids or corresponding control vectors. HA-IP (**A**,**B**,**D**,**E**) or FLAG-IP (**C**,**F**–**H**) products were prepared from the resulting cell lysates (input), as indicated. (**G**) Cells were co-transfected 1 d after treatment with siCtrl or LATS1 siRNA (siLATS1). (**A**–**H**) Transfected and/or endogenous proteins in input samples and IP products were detected using WB analysis with the indicated antibodies.

**Figure 5 ijms-24-14786-f005:**
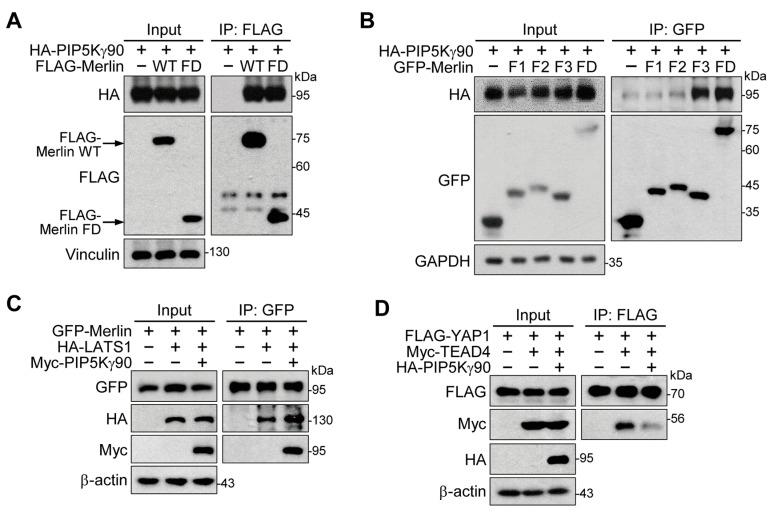
PIP5Kγ90 interaction with the Merlin FERM domain and its effects on interactions among Hippo signaling proteins. HEK293 cells were used for transfection and IP. (**A**) FLAG-tagged Merlin WT or FERM domain (FD) was co-transfected with HA-PIP5Kγ90. Arrows indicate FLAG-Merlin WT or FD. (**B**) GFP-Merlin FD or its F1, F2, or F3 subdomains were co-transfected with HA-PIP5Kγ90. GFP-Merlin (**C**) or FLAG-YAP1 (**D**) was co-transfected in the absence and presence of the indicated plasmids. Resulting cell lysates and FLAG-IP (**A**,**D**) or GFP-IP (**B**,**C**) products were analyzed using WB using the indicated antibodies.

**Figure 6 ijms-24-14786-f006:**
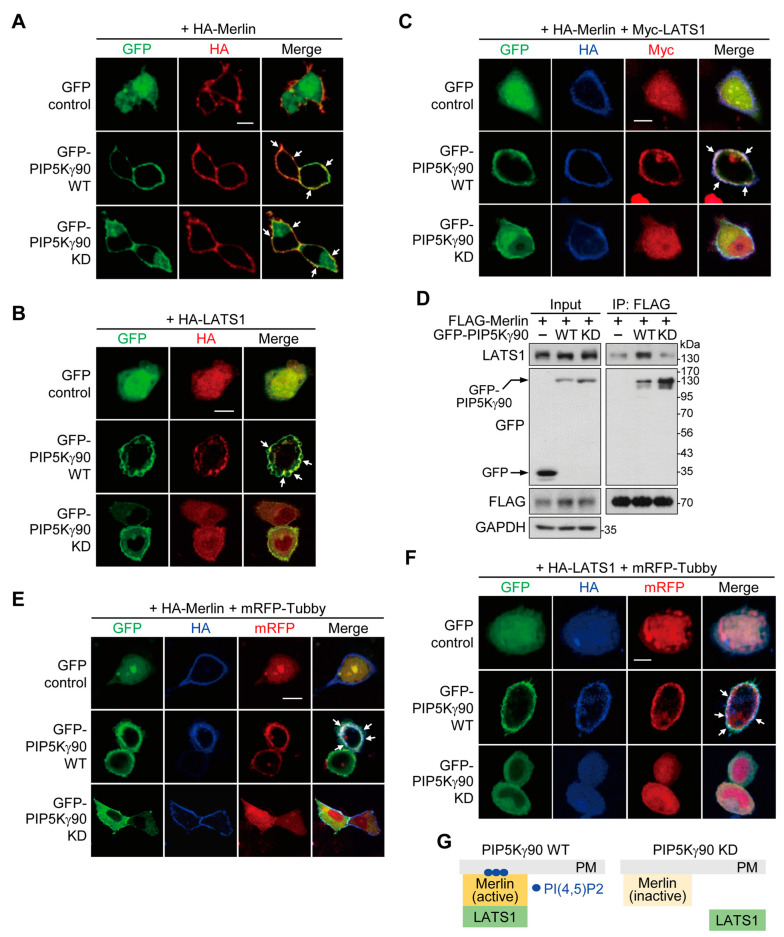
Effects of the PIP5Kγ90 WT and KD mutant on LATS1 co-localization with Merlin at the PM. (**A**–**C**,**E**,**F**) HeLa cells were co-transfected with GFP control vector, GFP-PIP5Kγ90 WT, or its KD mutant, together with HA-Merlin, HA-LATS1, Myc-LATS1, and/or mRFP-Tubby, as indicated. HA-Merlin, HA-LATS1, and/or Myc-LATS1 were immunostained using the respective primary antibodies followed by Alexa Fluor 594- and/or 350-labeled secondary antibodies. Representative images were captured using confocal microscopy. Arrows indicate enrichment of the transfected proteins at the PM. Scale bars, 10 μm. (**D**) HEK293 cells were co-transfected with GFP control vector or GFP- PIP5Kγ90 WT or its KD mutant with FLAG-Merlin. Cell lysates and FLAG-IP products were analyzed through WB using the indicated antibodies. Arrows indicate GFP-PIP5Kγ90 or GFP. (**G**) A schematic diagram of the results.

**Figure 7 ijms-24-14786-f007:**
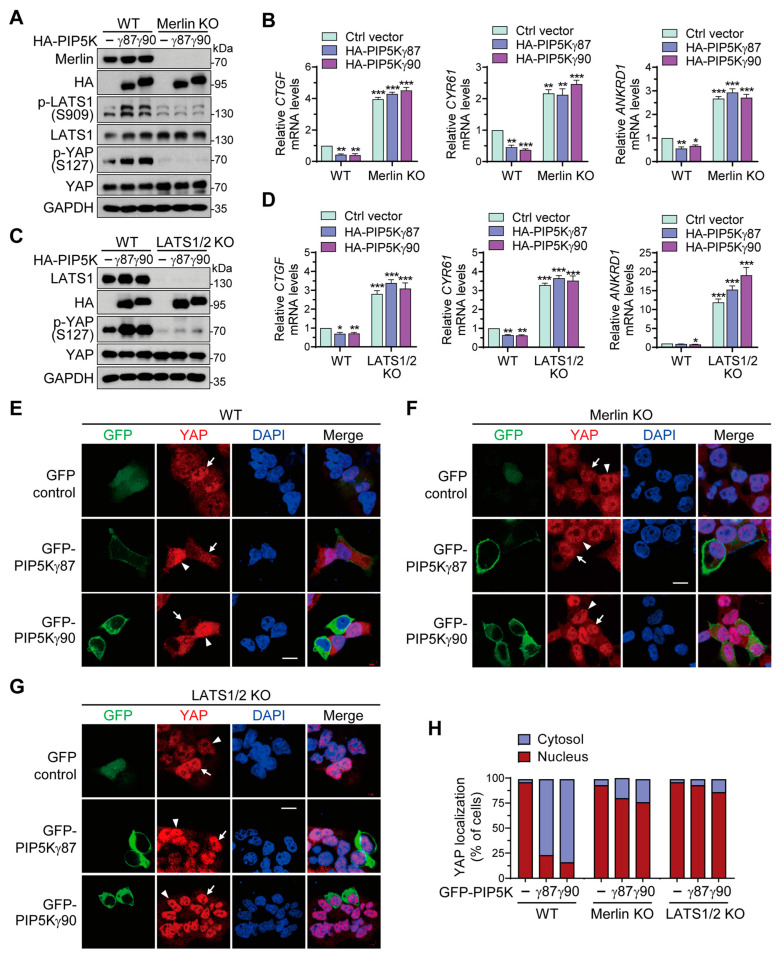
Effects of Merlin KO and LATS1/2 KO on Hippo pathway regulation by PIP5Kγ. WT and Merlin KO (**A**,**B**) or LATS1/2 KO (**C**,**D**) HEK293A cells were transfected with control vector, HA-PIP5Kγ87, or HA-PIP5Kγ90. (**A**,**C**) WB analysis of cell lysates with the indicated antibodies. (**B**,**D**) *CTGF*, *CYR61*, and *ANKRD1* mRNA levels as analyzed using qRT-PCR were quantified relative to those in control vector-transfected WT cells (*n* = 3). Values in the graphs represent the mean ± S.E.M. * *p* < 0.05, ** *p* < 0.01, *** *p* < 0.001. (**E**–**G**) WT, Merlin KO, or LATS1/2 KO HEK293A cells were transfected with GFP control vector, GFP-PIP5Kγ87, or GFP-PIP5Kγ90. YAP was immunostained using its primary antibody and Alexa Fluor 594-labeled secondary antibody; DAPI staining was used for nuclei visualization. Representative images were obtained using confocal microscopy. The arrows and arrowheads indicate transfected and non-transfected cells, respectively. Scale bars, 10 μm. (**H**) Relative abundance of cytosolic or nuclear YAP in GFP-positive cells (*n* = 30 each) collected from random fields in (**E**–**G**).

**Figure 8 ijms-24-14786-f008:**
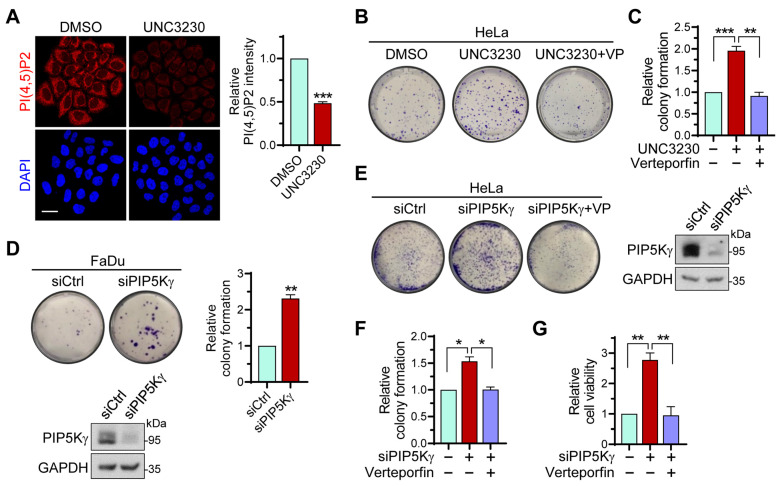
Effects of PIP5Kγ inhibitor or siRNA treatment on cancer cell proliferation. (**A**) Confocal images of PI(4,5)P2 from HeLa cells treated with dimethyl sulfoxide (DMSO) or the PIP5Kγ inhibitor UNC3230 (100 nM) for 3 d. Cells were identified using DAPI staining. PI(4,5)P2 fluorescence intensity was measured using ImageJ 1.53t software and relatively quantified. Scale bar, 20 μm. Colony formation and cell viability assays were performed as described in the Section 4. (**B**,**C**) Colony formation assay was performed using HeLa cells treated with UNC3230 and/or the YAP inhibitor verteporfin (VP, 1 μM), and the results were relatively quantified (*n* = 3). (**D**,**E**) Colony formation assay with FaDu and HeLa cells treated with control or PIP5Kγ siRNA in the absence or presence of verteporfin, as indicated. PIP5Kγ knockdown by siRNA was confirmed using WB analysis. (**D**) The number of cell colonies was relatively quantified using ImageJ software. (**F**) Colony formation assay results in (**E**) were relatively quantified (*n* = 3). (**G**) Cell viability was evaluated under the conditions as in (**E**) and quantified relative to the control condition (*n* = 4). (**C**,**F**,**G**) Values in the graphs represent the mean ± S.E.M. * *p* < 0.05, ** *p* < 0.01, *** *p* < 0.001.

**Figure 9 ijms-24-14786-f009:**
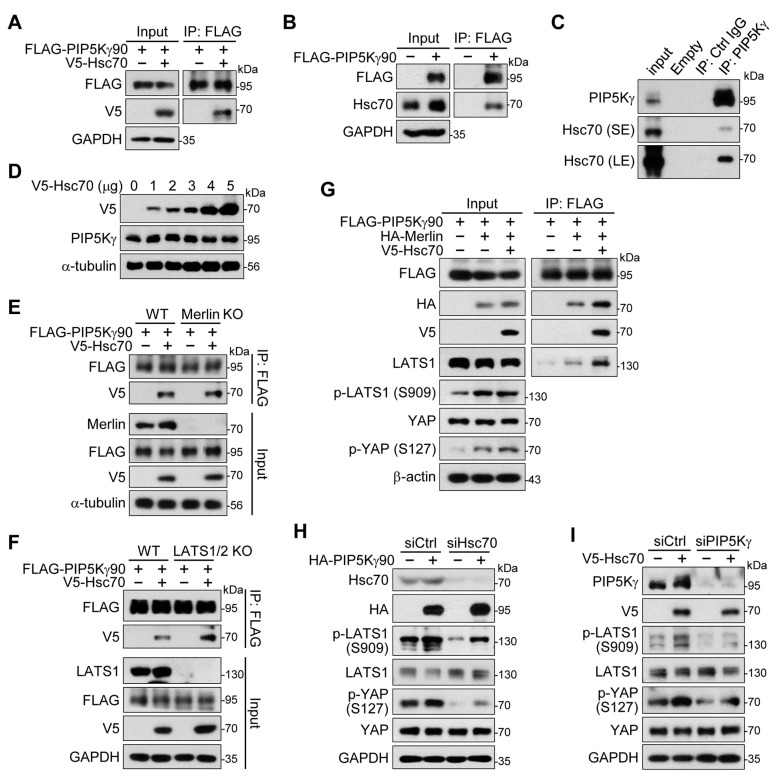
Hsc70 participates in Hippo–YAP pathway regulation by PIP5Kγ90. (**A**,**B**,**D**) Control vector, FLAG-PIP5Kγ90, and/or V5-Hsc70 were transfected into HEK293 cells, as indicated. (**C**) IP products using control IgG or PIP5Kγ antibody were prepared from HEK293 cell lysates. Hsc70 was immunoblotted with short-exposure (SE) and long-exposure (LE) times. WT, Merlin KO, and LATS1/2 KO HEK293A cells (**E**,**F**), HEK293 cells (**G**), Hsc70 siRNA (siHsc70)-treated HeLa cells (**H**), or PIP5Kγ siRNA (siPIP5Kγ)-treated HEK293 cells (**I**) were transfected in the absence and presence of the indicated plasmids. Resulting cell lysates (**A**–**I**) and FLAG IP products (**A**,**B**,**E**–**G**) were examined using WB analysis with the indicated antibodies.

**Figure 10 ijms-24-14786-f010:**
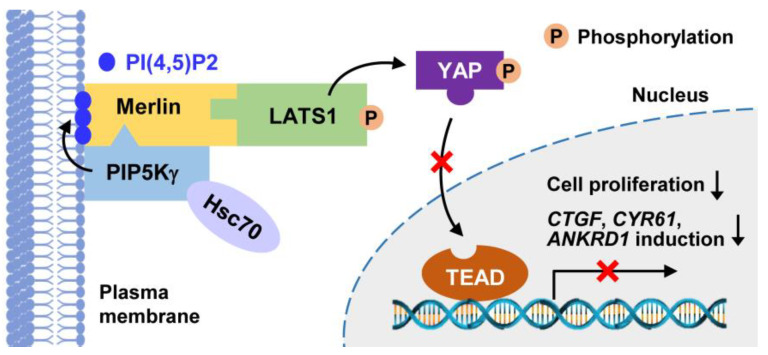
Proposed model for functional role of PI(4,5)P2-producing PIP5Kγ that regulates the Hippo–YAP signaling pathway by mediating a functional complex with Merlin and LATS1 in the plasma membrane, and via an interplay with Hsc70.

**Table 1 ijms-24-14786-t001:** qRT-PCR primer sequences.

Genes	Forward (5′–3′)	Reverse (5′–3′)
Human *CTGF*	CCTGCAGGCTAGAGAAGCAG	TGGAGATTTTGGGAGTACGG
Human *CYR61*	AAGAAACCCGGATTTGTGAG	GCTGCATTTCTTGCCCTTT
Human *ANKRD1*	TTTGGCAATTGTGGAGAAGTTA	AAACATCCAGGTTTCCTCCA
Human *GAPDH*	AGGGCTGCTTTTAACTCTGGT	CCCCACTTGATTTTGGAGGGA

## Data Availability

The data presented in this study can be made available on request from the corresponding author.

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
