# Peer review of "PIP5Kγ Mediates PI(4,5)P2/Merlin/LATS1 Signaling Activation and Interplays with Hsc70 in Hippo–YAP Pathway Regulation"

_ijms, 2023, doi:10.3390/ijms241914786_

Round 1

Reviewer 1 Report

The authors study the role of PIP5Kγ (Phosphatidylinositol 4-Phosphate 5-Kinase γ) in regulating the Hippo signaling pathway, which is essential for controlling tissue development and cell growth. Role of PIP5Kγ in the Hippo Pathway The authors show data on how PIP5Kγ is involved in the Hippo pathway. PIP5Kγ contributes to the activation of LATS1 and the inhibition of YAP through its production of PI(4,5)P2. PI(4,5)P2 induces conformational changes in Merlin, leading to its activation and facilitating interactions between Merlin, LATS1, and YAP. Overexpression of PIP5Kγ87 or PIP5Kγ90 increases the phosphorylation of LATS1 and YAP, leading to YAP inhibition. In contrast, PIP5Kγ knockdown reduces LATS1 and YAP phosphorylation levels, resulting in increased YAP activity. PIP5Kγ's role in the Hippo pathway has implications in cancer, as it can regulate cell proliferation. High levels of PIP5Kγ are associated with favourable prognosis in certain cancer types, and inhibition of PIP5Kγ can enhance cell proliferation. PIP5Kγ contains a motif that can bind to Hsc70, a heat shock protein. The interaction between PIP5Kγ and Hsc70 appears to play a role in Hippo pathway activation. PIP5Kγ affects the subcellular localization of Merlin and LATS1, particularly their recruitment to the plasma membrane, where they can interact and regulate the Hippo pathway. With these findings observed we may say that PIP5K alpha and PIP5K gamma play a analogous role in Hippo–YAP pathway regulation mediated by Merlin activation and PM recruitment of LATS1 in a PI(4,5)P2-dependent manner. PIP5K gamma only functions as an activator of Merlin rather recruiting it to plasma membrane. In summary, this study provides insights into how PIP5Kγ and PI(4,5)P2 production play a crucial role in regulating the Hippo pathway, affecting the phosphorylation and activity of key proteins involved in cell growth and tissue development. It also highlights the potential implications of PIP5Kγ in cancer and cell proliferation.  

Author Response

We do appreciate the Reviewer 1 for the evaluation of our study in a positive tone.

Reviewer 2 Report

In this manuscript, the authors explore the role of PIP5Kr in regulating the Hippo-YAP pathway. Firstly, the authors found that ectopic expression of PIP5Kr87 or PIP5Kr90 activated large tumor suppressor kinase 1 (LATS1) and inhibited Yes-associated protein (YAP), whereas PIP5Kr knockdown yielded opposite effects. Next, the authors further found that the regulatory effects of PIP5Kr were dependent on its catalytic activity and the presence of Merlin and LATS1. PIP5Kr knockdown weakened restoration of YAP phosphorylation upon stimulation with epidermal growth factor or lysophosphatidic acid. The authors also found that PIP5Kr90 bound to the Merlin’s FERM domain, forming a complex with PIP2 and LATS1 at the PM, and PIP5Kr90, but not its kinase-deficient mutant, potentiated Merlin–LATS1 interaction and recruited LATS1 to the PM. In addition, the authors also found that PIP5Kr90 interacted with heat shock cognate 71-kDa protein (Hsc70), which also contributed to Hippo pathway activation. In general, the authors present lots of experimental results, and my assessment is positive. However, I think some minor issues should be addressed prior to publication in IJMS.

Specific comments:

1.    In figure 1, 2 and 7, I suggest the authors quantify the western blot bands of p-LATS1 and p-YAP. In addition, whether the authors can test the protein expression of CTGF, CYR61 and ANKRD1?

2.    In figure 3B, whether the authors quantify the foci number or fluorescence intensity of GFP-PLC-PH in the PM?

3.     In figure 8A, whether the authors can quantify the fluorescence intensity of PIP2?  In addition, in figure 8D, whether the authors can count the colony number?

4.    In figure 9H-I, I suggest the authors quantify the western blot bands of p-LATS1 and p-YAP.

Reviewer 3 Report

In this study, the authors investigated the lipid kinase PIP5K1C (PIP5Kgamma) in regulating the Hippo-YAP activity. The same group recently just published a very similar study for PIP5K1A. Both kinases have been known to regulate the Hippo-YAP activity.

The current study provides large amount of data and also provides some new insights regarding how this kinase regulates the Hippo-YAP signaling.

1) in Fig. 3C, it seems YAP S127 level was not decreased in siRNA PIP5Kg samples. based on the context, it was expected to be lower like in Panel A.

2) the authors showed the expression levels of the kinases in clinical samples (in supplemental materials), but they did not do correlation analysis, which might further strengthen the connection between these kinases/lipids and YAP in the clinic. Is there any correlation between YAP and PIP2/PIP5K in cancer patient samples?
